psychology

perceptual fluency, inhibition, preference, learning, memory

**Author for correspondence:**
Jonathan C. Flavell
e-mail: jonathan.flavell@york.ac.uk

# Three minutes to change preferences: perceptual fluency and response inhibition

Bryony McKean, Jonathan C. Flavell, Harriet Over and Steven P. Tipper

Department of Psychology, University of York, United Kingdom

 JCF, 0000-0001-9521-8827; HO, 0000-0001-9461-043X

Perceptual fluency and response inhibition are well-established techniques to unobtrusively manipulate preference: objects are devalued following association with disfluency or inhibition. These approaches to preference change are extensively studied individually, but there is less research examining the impact of combining the two techniques in a single intervention. In short (3 min) game-like tasks, we examine the preference and memory effects of perceptual fluency and inhibition individually, and then the cumulative effects of combining the two techniques. The first experiment confirmed that perceptual fluency and inhibition techniques influence immediate preference judgements but, somewhat surprisingly, combining these techniques did not lead to greater effects than either technique alone. The second experiment replicated the first but with changes to much more closely imitate a real-world application: measuring preference after 20 min of unrelated intervening tasks, modifying the retrieval context via room change, and generalization from computer images of objects to real-world versions of those objects. Here, the individual effects of perceptual fluency and inhibition were no longer detected, whereas combining these techniques resulted in preference change. These results demonstrate the potential of short video games as a means of influencing behaviour, such as food choices to improve health and well-being.

## 1. Introduction

The ability to change an individual's behaviour has significant potential for many of the issues of the day, from changes in lifestyle to reduce energy consumption [1] and improved diet [2] to reducing interpersonal conflict [3]. One route to such behaviour change is to influence preference and choice, where

an increase in preference for stimuli such as healthier foods can lead to increased healthy food consumption and improvements in health. We approach this issue by considering the basic visual-motor processes that could be potentially manipulated while playing computer games.

Games to change behaviour have proliferated in recent years. In particular, brain training games purported to improve cognitive processes have been widely promoted. However, there has been much debate concerning whether such games can change behaviour in a broad and general manner. A meta-analytic review by Sala *et al.* [4] suggests changes are domain specific and rarely generalize to wider situations. Similarly, Brox *et al.* [5], in reviewing video games designed to improve health, concluded that there is not enough evidence to decide which design principles are most effective, as there is a limited specification of design methodologies.

Therefore, in order to understand how these might influence preference changes, there is a need for basic research to examine the core processes of perception and action. Once identified, these very basic processes can be embedded in engaging game environments as a means of implicitly shifting preference towards or away from given stimuli such as healthy and unhealthy foods. As we will see below, there is evidence that some particular core processes, such as inhibition, are able to shift preferences. However, an important further issue is to examine different processes (such as perceptual fluency and inhibition) in the same experimental context (see also [6,7]). And, more importantly, it is necessary to investigate whether combining various perception-action processes is a more effective way of modifying preference than targeting any one of those processes in isolation in more demanding learning conditions. To address this, we examined perceptual fluency and inhibition independently and in combination in the same task environment.

## 1.1. Perceptual fluency

Perceptual fluency is the subjective feeling of ease or difficulty while processing perceptual information [8]. A substantial body of research has now confirmed that more fluent processing results in positive assessments of perceptual stimuli. The first such demonstration was the mere exposure effect [9] where the repeated presentation of a stimulus resulted in more fluent processing and hence greater liking. Priming effects, increased contrast and readability have also produced increased preference [10], increase in trust [11] and decrease in associated risk [12]. Similarly, object symmetry [13–15] and round versus sharp edges [16] have produced clear preference effects, and even influenced taste judgements [17].

However, an important issue concerning the applicability of perceptual fluency to real-world behaviour change remains to be investigated. That is, although the effects are robust and observed across a range of perceptual features, most are observed during stimulus processing. For example, a stimulus with higher contrast is preferred or trusted more while participants are directly viewing the stimuli [10,18,19]. Critically, however, effects must be stable across time and generalize to other non-laboratory-task contexts for perceptual fluency effects to influence real-world behaviours. For example, producing a preference for healthy foods via perceptual fluency while playing a computer game is of little value if this preference does not generalize beyond the task context. As noted above, this has been a central concern for a range of gamification applications, such as the lack of generalization from brain training to non-game contexts [4].

This contrast between fluency effects while directly engaged with a stimulus, and the lack of effects when assessing preference at a later time, was noted by Cannon *et al.* [20]. In that study, electromyographic recordings of facial muscles revealed emotional response to fluent versus disfluent action towards a stimulus. However, at later assessments of object liking where particular object identities had been repeatedly associated with either fluent or disfluent actions, no such preference effects were observed. Similarly, Flavell *et al.* [18] presented participants with objects that consistently moved fluently or disfluently. They demonstrated that the motion fluency influenced object liking while the objects were assessed while moving, but there were no effects when the objects were subsequently assessed while stationary. Hence there appeared to be little associative learning between the consistent pairing of object-identity and its motion (see also Strachan *et al.* [21] for similar findings in judgements of face trustworthiness). These are important boundary conditions that might limit the utilization of perceptual fluency to applied issues of preference and choice, and hence they require further research.

## 1.2. Inhibitory processes

In parallel to the above-described studies of perceptual fluency, other research has investigated the role of inhibition in changing preferences. A range of studies have demonstrated that the liking of a stimulus can

be reduced by: inhibition associated with ignoring the stimulus (e.g. [22]), inhibition of return following exogenous attention cues associated with the stimulus (e.g. [23]) and more explicit stop-signal inhibition (e.g. [24]). The role of inhibition in changing food preferences has been particularly interesting, where a range of studies demonstrate that inhibition (stop-signal and go/no-go techniques) can be associated with undesirable/unhealthy foods leading to reduced consumption later (see [25], and see Veling et al. [26] for review).

Of importance, and in contrast with perceptual fluency effects, inhibition training appears to be quite robust, especially for participants who are frequently dieting [27] and the effects generalize from the laboratory task to the actual consumption of foods at a later time [28]. These inhibition training and evaluative conditioning processes, where there is an association of inhibition with particular stimuli such as foods is of importance for practical application. It is of note, however, that the experiments employing inhibition to change subsequent preference and choice tend to require participants to undertake several hundred trials (e.g. [29,30]). Hence, it is not clear whether inhibition is a more effective means of changing preference than perceptual fluency, or whether the former employs more trials to produce more stable learning.

Therefore, in the current experiments, we investigate both perceptual fluency and inhibition in essentially the same experiment, which enables an initial examination of the two processes. Furthermore, we examine whether preference can be shifted in a very short 3 min game-like task with only 16 trials in each of the key preference changing experimental conditions. As far as we are aware, this is a significantly shorter procedure than typically employed (e.g. five blocks of 24 trials were used by Veling et al. [27]). If effects can be detected in such a short period of time, then this bodes well for such techniques in games where repeated returns are the norm, where spaced learning has been shown to result in more robust learning and retrieval (e.g. [31]).

This approach enables a further investigation of a potentially important issue. Thus far, research tends to focus on one approach, such as studies examining a particular inhibition approach such as stop-signal, or perceptual fluency approaches such as the effect of contrast or symmetry. Of course, when using techniques in a game setting where players interact with characters to achieve goals over multiple levels and repeated playing episodes, a wide range of techniques could be integrated. For example, to reduce preference for unhealthy foods, inhibiting action towards those stimuli on some trials could be merged with various perceptual disfluency techniques on other trials. Embedding in a game the wide range of established perceptual fluency and inhibition techniques could result in deeper encoding and more stable longer term effects that generalize beyond the game. Of particular note, we currently have limited knowledge of how combining techniques, such as inhibition and perceptual fluency with very few stimulus exposures, affects the stability of preference change. We intend to explore this issue for the first time in the current series of simple short game-like tasks.

# 2. Experiment 1

The task is game-like, in that participants are required to 'catch' rapidly disappearing food items (grapes) by reaching out and touching them on a touch screen. This study directly compares three manipulations in a between-subjects fashion: a perceptual fluency condition (as stimulus presentation rate); an inhibition condition (as stop-signal task e.g. [2]) and a combination of the perceptual fluency and inhibition manipulations into a single condition.

It is important to note that these are initial exploratory experiments. Unlike the work of van Koningsbruggen et al. [28] and Lawrence et al. [32] for example, at this stage we are not attempting to tackle real-world issues, such as shifting preference from unhealthy foods (e.g. pizza, crisps, chocolate) towards healthier fruit and vegetables. Although that is a critical long-term goal of our work, these initial basic experiments investigate shifts between similar stimuli in the same category. This less demanding preference change would seem to be the appropriate initial approach to examine whether preference can indeed be changed with a very short task lasting just 3 min, and to initially directly investigate perceptual fluency, inhibition and the combination of these techniques.

## 2.1. Method

### 2.1.1. Apparatus

Participants sat at a table in a lit room facing a 23″ touch-screen monitor (HannsG (Taipei, Taiwan) HT231HPB, 1920 × 1080 pixels). A keyboard was positioned on the table between the participant and

**Figure 1.** (a) Experiment 1 stimuli. (b) Experiment 1 and 2 possible stimuli locations (+) and stop-signal location (○, inhibition and combined conditions only). (c) Stop-signal (yucky face) stimulus. (d) Experiment 2 stimuli. (e) Schematic of the task on a grape trial in perception condition. From left to right: instructions to hold space bar; participant presses causing objects to appear; participant releases the space bar and reaches to touch one of the grape objects.

the screen approximately 30 cm away. Participants and the keyboard spacebar were positioned at the screen's horizontal centre. Stimulus presentation (60 Hz) and response recording were achieved using custom scripts and Psychtoolbox 3.0.11 [33–35] operating within Matlab 2018a (The MathWorks Inc., Natick, USA) on a PC (Dell (Round Rock, USA) XPS, Intel® Core™ i5-4430, 3 GHz CPU, 12 GB RAM, 64 bit Windows 7).

### 2.1.2. Experiment composition

The experiment consisted of a practice block, a task block and a rating block in that order. Participants were assigned to either a perception, inhibition or combined (perception + inhibition) condition. The practice and task blocks differed between conditions but the rating block was always the same. Fluency was manipulated using presentation time (fast/slow) in the perception condition and stop-signal association (present/absent) in the inhibition condition. This led to a $3 \times 2$ between-within design.

### 2.1.3. Perception condition trials

At the start of a trial, participants pressed and held the space bar with the index finger of their dominant hand. This caused 10 objects to appear on the screen simultaneously. The objects were either all green grapes, all red grapes, all green leaves or all red leaves, (figure 1a). Each object was randomly located on a grid (figure 1b). The objects would then disappear one at a time.

Participants were instructed that if grapes appeared then they should release the space bar and, using the same finger, reach out and touch one of them (figure 1e), but if leaves appeared, they should keep the space bar held down until the trial ended. This task requirement of responding to grapes and not leaves ensured participants actively identified the grapes. Participants were told to keep attempting to touch grapes until they were either successful or until the trial ended when all the grapes had disappeared. A trial ended when either a grape was touched or when all of the objects had disappeared.

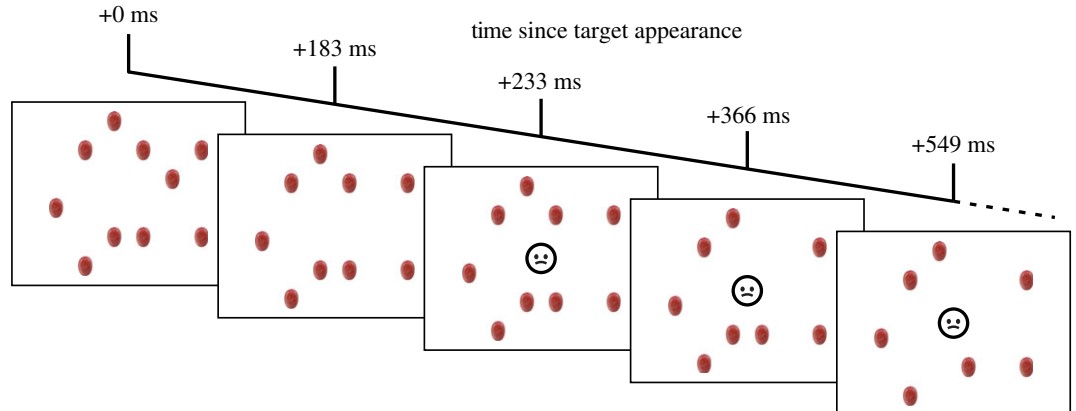

**Figure 2.** Schematic of the target disappearance and stop-signal appearance in the inhibition condition. Timings in the figure are relative to the initial appearance of the targets. From left to right: 10 grapes appear on screen; 183 ms later one grape vanishes; 50 ms later the stop-signal appears; 133 ms later one more grape vanishes (i.e. 183 after the first grape vanished); 183 ms later one grape vanishes. Grapes continued to disappear one at a time every 183 ms until the trial ended.

Instructions were given verbally before the practice block and presented onscreen (see osf.io/vxbzj/ for verbatim copies).

Touching a grape or not releasing the spacebar for leaves resulted in a successful trial. Failing to touch a grape before they all disappeared or releasing the spacebar for leaves resulted in an unsuccessful trial. After a successful trial, the next trial would begin immediately, whereas after an unsuccessful trial an error tone would play and all on-screen objects would be replaced by feedback reading either 'OH NO, you shouldn't have saved the leaves' or 'OH NO, you didn't save any grapes' as appropriate.

One colour of grape would disappear faster than the other (counterbalanced between participants). The fluent grapes (which disappeared slowly and so were easier to catch) disappeared at a rate of one every 233 ms and the disfluent grapes (which disappeared more quickly and so were more difficult to catch) disappeared at a rate of one every 100 ms. Both colours of leaf disappeared at a rate of one every 183 ms. Participants were not told of the differences in disappearance rates.

### 2.1.4. Inhibition condition trials

These trials were exactly the same as the perception condition with two changes: the inclusion of a stop-signal and the rate of object disappearance.

In addition to the perception instructions, participants were told to not respond if a yucky face (the stop-signal, figure 1c and figure 2) appeared during a trial.[1] The stop-signal face could only appear in the centre of the screen and would remain until the trial ended. If they responded when the face appeared, an error tone would play and all on-screen objects would be replaced by feedback reading 'OH NO, you pressed at a yucky face'. The face would only ever appear for one colour of grape (red or green, counterbalanced across participants, see condition assignments) and would never appear on leaf trials. The face would appear on 50% of the assigned grape's trials.[2]

All grapes disappeared at a rate of one every 183 ms and the yucky face appeared 50 ms after the first grape disappeared (i.e. 233 ms after the grapes appeared). The leaves also disappeared at a rate of one every 183 ms.

### 2.1.5. Combined condition trials

These trials were a combination of the perception and inhibition conditions. The fluent (slow) grapes disappeared at a rate of one every 233 ms and were never paired with the yucky face. The disfluent

---

[1]The yucky face stimulus was used as the stop-signal with an eye to our future research goals. That is, this basic research is providing the foundation for our game development ideas, and the target of such healthy eating interventions will be children. Hence, this child-like stimulus may be more effective. However, we acknowledge that the stimulus does have an emotional property, and hence there could also be some evaluative conditioning component (e.g. [36]).

[2]Previous research has suggested that go/no-go procedures might be more effective than stop-signal techniques (e.g. Allom et al [37]). However, in the current experiments, stop-signal on 50% of trials is used rather than other procedures, because this 50% stop-signal approach enables the combination of perceptual fluency and inhibition within the same displays. That is, it is necessary for response to be produced on some trials for the perceptual-fluency technique to be employed when techniques are combined.

(fast) grapes disappeared at a rate of one every 100 ms and were also paired with the yucky face on 50% of trials. Participant instructions and remaining details were identical to the inhibition condition.

### 2.1.6. Rating trials

After playing the catch the grape game, a single red or green grape would be presented in the screen centre with instructions to rate it between 1 (dislike) and 9 (like) using the keyboard's number pad. The other colour of grape would then be presented for liking rating.

### 2.1.7. Block composition

In all conditions: the practice block consisted of four grape trials and two leaf trials; the task block consisted of 32 grape trials and 16 leaf trials; and rating block consisted of two grape trials. In each block, half of grape trials were fluent and half were disfluent, and half of leaf trials were green, and half were red. Trial order was randomized within blocks between participants. The experiment was self-paced after verbal instruction by the experimenter at the start of the experiment. The practice and task blocks took approximately 3 min to complete in all conditions.

### 2.1.8. Stimuli

The red grape was a re-coloured version of the green grape. A single leaf was re-coloured to closely match the two grape colours. Leaf outlines were edited to match the profiles of the grapes. Edits made with CorelDraw 2018 v. 20 (Corel Corporation, Ottawa, Canada).

### 2.1.9. Data exclusion and analysis

Participants were excluded if they responded on more than 25% of leaf trials in the task block. Statistical analysis was conducted in JASP v. 0.10.1.0 [38]. We took a Bayesian approach to analysis and report the evidence categories ('no evidence', 'anecdotal', 'strong', 'very strong', etc) as described in Wagenmakers *et al.* [39]. Full Bayesian models and frequentist alternatives are listed on osf.io/vxbzj/.

### 2.1.10. Participants

All participants were recruited from the University of York's Department of Psychology participant recruitment system. Participants received either course credit or financial compensation for participation. Protocols were approved by the University of York's Psychology Departmental Ethics Committee and were in accord with the tenets of the Declaration of Helsinki. Participants gave written consent but were naive to the purpose of the research until participation was complete.

In the perception condition, 45 participants were tested. Five participants were above the error threshold giving a final sample of 40 participants (4 males, age mean ± s.d. = 22.7 ± 9.1) with none responding on more than 4 of 16 leaf trials (mean ± s.d. = 1.7 ± 1.2). In the inhibition condition, 42 participants were tested. Two participants were above the error threshold giving a final sample of 40 participants (8 males, age mean ± s.d. = 19.8 ± 1.4) with none responding on more than 3 of 16 leaf trials (mean ± s.d. = 0.6 ± 0.9). In the combined condition, 41 participants were tested. One participant was removed from the analysis for failing to complete the experiment giving a final sample of 40 participants (4 males, age mean ± s.d. = 20.4 ± 1.8) with responding on more than 3 of 16 leaf trials (mean ± s.d. = 0.9 ± 0.9).

## 2.2. Results

### 2.2.1. Task performance

Error rates for grape trials in each condition are reported in osf.io/vxbzj/.

### 2.2.2. Rating data

Liking ratings for fluent/disfluent grapes are shown in figure 3, left panel. A $3 \times 2$ (condition × fluency) Bayesian repeated-measures ANOVA on liking ratings most strongly supported a model including only the fluency term ($BF_{10} = 3.64 \times 10^{11}$, $p(H_1 \mid Data) = 0.912$: fluency $BF_{incl.} = 2.485 \times 10^{11}$). Fluent grapes were

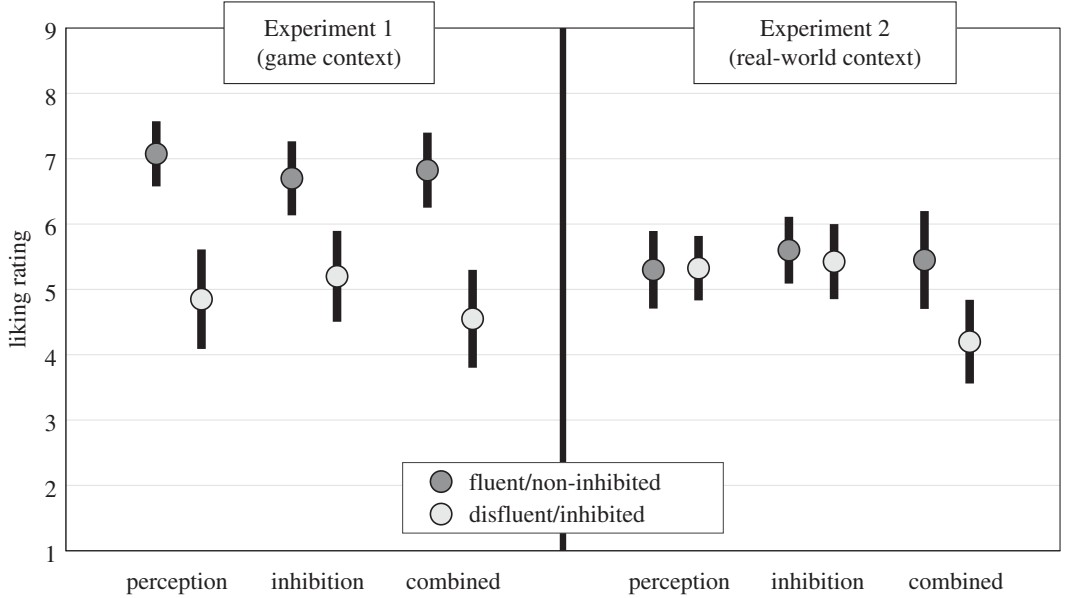

**Figure 3.** Mean (±95 confidence interval) liking ratings of grape objects (Experiment 1, left panel) and taste ratings of drinks (Experiment 2, right panel) in each fluency × condition manipulation.

preferred over disfluent grapes regardless of the conditions in which they were experienced. This indicates that our perceptual, inhibition and combined manipulations are either equally potent preference shifters, or that each are sufficiently potent to reach ceiling effects on preference shifting.

## 2.3. Discussion

The results of this first experiment are clear. We have demonstrated that both perceptual fluency and inhibition techniques can rapidly shift preference within our 3 min task. This confirms the efficacy of inhibition as a technique for influencing preference decisions. In terms of perceptual fluency, this is one of the first demonstrations of preference effects when not directly observing fluent or disfluent stimuli. That is, ratings were not taken when the stimuli were disappearing from the screen at different rates, but at the end of the experiment, when presented statically for some seconds. This result is in contrast with that of Flavell *et al.* [18] who showed no transfer from moving to static stimuli rated at the end of the experiment. Hence, it would appear that perceptual fluency may be similar to inhibition, in that effects transfer beyond in-task experiences.

The second main observation is the somewhat counterintuitive finding concerning the combined perception + inhibition condition. Our initial hypothesis was that combining the perceptual fluency and inhibition would produce more robust preference changes than either alone. This clearly was not the case. It is possible that our effects in the current experiment are at the ceiling. That is, the preference rating task is a few seconds after the completion of the experimental game task, and hence even though combined techniques might indeed be more robust, that cannot be detected in these short-interval conditions. Hence, we hypothesize that more demanding memory retrieval might reveal benefits of combined techniques.

## 3. Experiment 2

The game-like task in Experiment 2 is similar to that of Experiment 1. However, unlike the usual experimental approach of changing just one variable such as stimulus type, we changed a range of experimental properties to assess possible eventual real-world impact. That is, Experiment 2 is a much more severe challenge of learning and memory retrieval. This is of fundamental importance if techniques embedded in games are to have real-world effects on preference and choice. Therefore, in Experiment 2, a number of significant changes are made following the completion of the task/game. First, after completing the 3 min game participants engage with a completely different task for

approximately 20 min, rather than immediately making a preference judgement. This task takes place in a new room, to change the environmental context [40,41]. After completing the new task, participants then move in to yet another new room, further increasing changes in an environmental context. Finally, and of perhaps most importance, preference judgements are made with real-world stimuli. That is, they are presented with drinks to taste and make preference judgements on. For games to be effective, a crucial step is for transfer from the images of objects on the computer screen to real-world objects that can be consumed.

## 3.1. Method

The experimental procedure was exactly as described for Experiment 1 apart from where stated below.

### 3.1.1. Practice, task, rating and interview blocks

Before the practice block, participants were told that they would complete several unrelated experiments in different rooms. Participants completed the practice and task blocks as described in Experiment 1, but the object images were changed from grapes and leaves to drinks in cups to be reached and touched and outlines of cups to-be-ignored (figure 1d and Stimuli). The instructions were changed to match these new terms. The practice and task block took place in the same room as in Experiment 1. After completing these blocks, participants were taken to a second room where they completed an unrelated visual search task for approximately 20 min. They were then taken to a third room where they completed a surprise rating block.

The rating block was completed in a lit room at a white desk with a white backdrop. On the desk were two cups of coloured water (see Stimuli) set approximately 30 cm back from the seating edge and approximately 50 mm separate from each other. Whether the fluent (i.e. slow disappearing and/or stop-signal association) drink was on the left or right was counterbalanced within conditions. In front of each drink, was a paper 9-point Likert scale, and between these was a ballpoint pen. Participants were seated centrally in front of the drinks and asked to try both the drinks, drink as much as they liked and rate how much they liked the taste of each. The researcher then left the room while ratings were made but surreptitiously waited to see which drink was picked up first. In addition to the Likert scale rating of each drink, we also analysed which drink was picked up first and the volume consumed of each drink.

After completing the rating task, participants completed a surprise funnelled interview to explore whether they explicitly noticed the experimental manipulation. All participants were first asked an open-ended question 'Did you notice anything about the game you played with the drinks and the cups?'. The follow-up question then depended on condition. In the perception condition, participants were asked 'One of the drinks disappeared quicker than the other, was this the green or blue drink?'. In the inhibition condition, participants were asked 'One of the drinks was always paired with the yucky face, was this the green or blue drink?'. In the combined condition participants were asked both of those questions.

### 3.1.2. Stimuli

When transferring to real-world object assessments, for health and safety reasons we substituted green and red grapes for cups of green- and blue-dyed water. Green and red leaves were substituted for green and blue outlines of those cups so that they presented the same profile. The cups were transparent plastic cups filled with 400 ml of water. Colour was controlled using set volumes of food dye which were determined before data collection so that both drinks would be similarly vivid yet transparent. These drinks were photographed from the same perspective for use as stimuli in the computer game. The outline images were closely colour matched to the corresponding liquid colour using CorelDraw 2018 v. 20 (Corel Corporation, Ottawa, Canada). See figure 1d for reference.

As in Experiment 1, participants rated only the test objects in the rating block. These drinks were prepared in exactly the same way as those photographed for the practice and task blocks.

### 3.1.3. Participants

In the perception condition, 43 participants were tested. One participant exceeded the error threshold and two participants did not complete the taste test giving a final sample of 40 participants (4 males, 5 did not

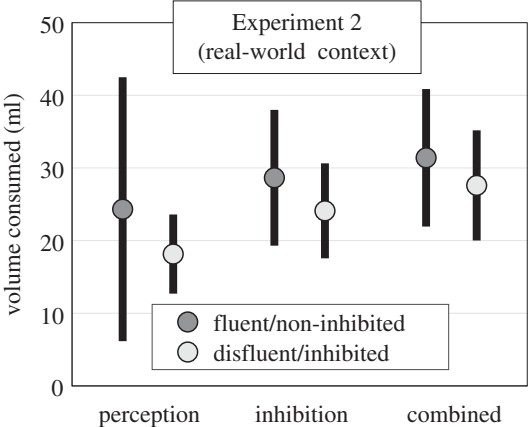

**Figure 4.** Mean (±95 confidence interval) of volume consumed in each fluency × condition manipulation of Experiment 2.

disclose gender, age mean ± s.d. = 19.9 ± 3.4) with none responding on more than 3 of 16 cup trials (mean ± s.d. = 1.1 ± 0.9). In the inhibition condition, 41 participants were tested. One participant did not complete the taste test giving a final sample of 40 participants (8 male, 5 did not disclose, age mean ± s.d. = 21.0 ± 6.8) with none responding on more than 3 of 16 cup trials (mean ± s.d. = 0.4 ± 0.7). In the combined condition, 41 participants were tested, one participant did not complete the taste test giving a final sample of 40 participants (5 male, 5 did not disclose, age mean ± s.d. = 21.9 ± 15.8) with none responding on more than 3 of 16 cup trials (mean ± s.d. = 0.7 ± 0.9).

## 3.2. Results

### 3.2.1. Task performance

Error rates for drink trials in each condition are reported in osf.io/vxbzj/.

### 3.2.2. Taste ratings

Taste ratings for fluent/disfluent drinks are shown in the right panel of figure 3. A 3 × 2 (condition × fluency) Bayesian repeated-measures ANOVA on liking ratings most strongly supported a model including the fluency, condition and fluency × condition terms ($BF_{10}$ = 7.113, $p(H_1 | Data)$ = 0.554: fluency $BF_{incl.}$ = 5.278; condition $BF_{incl.}$ = 1.522; fluency × condition $BF_{incl.}$ = 4.965). To break down the interaction, we performed Bayesian two-tailed paired samples $t$-tests on each of the perception, inhibition and combined conditions. Fluency effects were supported in the combined condition ($BF_{10}$ = 17.334), but there was evidence against such effects in the inhibition ($BF_{10}$ = 0.215) and perception ($BF_{10}$ = 0.171) conditions.

### 3.2.3. Drink picked first

Note that several data points were missing where it was not possible for the experimenter to surreptitiously take note of which drink was chosen first. Bayesian binomial tests (test value = 0.5) indicated that there was moderate evidence that participants were not above chance at choosing the perceptually fluent (19 of 37 participants, 51.4%, $BF_{0+}$ = 4.33) or non-inhibited (18 of 39 participants, 53.8%, $BF_{0+}$ = 3.326) drinks first. However, participants were above chance at choosing the fluent/non-inhibited drink first in the combined condition (25 of 39 participants, 64.1%), though this was marginal ($BF_{+0}$ = 1.749).

### 3.2.4. Volume consumed

A 3 × 2 (condition × fluency) Bayesian repeated-measures ANOVA on the volume drunk did not support any model above the null (see osf.io/vxbzj/ for models). That is, there was no evidence for differences in volume consumed in all models. See figure 4 for reference.

### 3.2.5. Funnelled interview (explicit awareness)

Responses to interview questions were analysed using Bayesian binomial tests (test value = 0.5).

In the perception condition, there was anecdotal evidence that less than half of participants correctly reported the perceptual manipulation (rate of drink disappearance) in the open-ended question (37.5% correct, $BF_{-0} = 1.255$). Of the participants who did not correctly identify the manipulation ($n = 25$), there was moderate evidence that participants were not above chance at correctly identifying which drink colour disappeared faster in the subsequent explicit question (48% correct, $BF_{0+} = 4.769$).

In the inhibition condition, there was very strong evidence that less than half of participants correctly reported the inhibition manipulation (face only appearing for one drink colour) in the open-ended question (25% correct, $BF_{-0} = 63.228$). Of the participants who did not correctly identify the manipulation (data for one participant were missing from this subset meaning only 29 participants were analysed), there was moderate evidence that participants were not above chance at correctly identifying which drink colour was paired with the yucky face (51.7% correct, $BF_{0+} = 3.787$).

In the combined perception + inhibition condition, there was moderate evidence that less than half of participants correctly reported the perception and inhibition manipulations in the open-ended question (32.5% correct, $BF_{-0} = 4.396$). Of the participants who did not initially correctly identify the manipulations ($n = 27$), there was moderate evidence that participants were not above chance at correctly identifying which drink colour was paired with the yucky face (48.1% correct, $BF_{0+} = 4.92$), and there was moderate evidence that participants were not above chance at correctly identifying which drink colour disappeared faster (37% correct, $BF_{0+} = 9.517$).

In summary, there was no evidence that participants were above chance at identifying the experimental manipulations in any condition.

## 3.3. Discussion

The results of Experiment 2 are in sharp contrast to those of Experiment 1. Recall that in Experiment 1, when preference judgements were made immediately after completing the experiment/game, clear effects of perceptual fluency and inhibition were observed, and there was no advantage when these techniques were combined. However, the effects dramatically change in Experiment 2, which features more severe memory challenges of longer task-filled intervals, a change of environmental context and transfer from on-screen images to real-world objects. Now the previously robust perceptual fluency and inhibition effects on preference are no longer detected. Although we cannot identify which components, such as stimulus change or longer retention interval, might be more disruptive, the main goal was to examine the effects in more real-world situations. Hence, although this short 3 min task can in principle alter preference, the effects of the individual techniques are fragile. However, we now discover that when these techniques are combined within the same block of trials, preference change is indeed more robust, generalizing to real-world challenges.

Examining preference effects while interacting with real-world objects such as drinks enabled us to measure two other possible preference changes via implicit measures that did not require participants to introspectively interrogate their preferences. The first of these was the amount of drink consumed, where it might be hypothesized that more of the preferred drink would be consumed. However, we did not detect any differences between drinks previously associated with fluent or disfluent processing. One possible reason we did not detect this consumption difference is due to floor effects. That is, the drinks were tap water that was not very appealing, and only 26 ml was consumed on average, which equates to a couple of small sips. On the other hand, our second implicit measure of which drink was picked up first did support our explicit preference rating results. That is, no differences were detected for the individual perception and inhibition conditions, but there was evidence for a greater proportion of participants picking up the fluent/uninhibited drink first in the combined perception + inhibition condition. That is, after drinks were associated with both disfluent perceptual processes (faster presentations) and stop-signal inhibition, participants were less likely to taste them first.

As a final comment, which must be tentative at this stage, the majority of participants appear to have little knowledge of the game properties that might have influenced their later preference decisions. That is, the majority of people could not report anything about the perceptual and/or inhibition features of the game. And even when provided information about what these game properties were, they still could not explicitly identify which drink was associated with disfluent processing. Although the issue of awareness requires much more work, it is intriguing that the visuomotor processes that might be embedded in games might be able to change preferences without awareness.

# 4. Conclusion

A central goal of this research was to examine whether preference can be changed in very short periods. The task described here was completed in approximately 3 min and each fluent or disfluent stimulus was only presented 16 times. This is a very challenging test of the perceptual fluency and inhibition mechanisms. Somewhat surprisingly, both perceptual fluency and inhibition do change preference very rapidly, if the assessment is made immediately after completing the game. For more stable longer term effects that generalize to new contexts and stimuli, we have now demonstrated that combining techniques is more effective.

The combination of mechanisms has implications for the greatest challenge for gamification approaches. That is, the challenge is for the processes within the game environment to leave a trace in the memory and shift behaviour and performance in the wider real-world context beyond the game. It is one thing to bias food preference in a child while playing a game, and quite another to bias choice towards healthier options at the dinner table at a later time. As noted, explicit inhibition training techniques do appear to have achieved this transfer from computer task to real-world food consumption, at least with long and repeated training sessions, as made clear in recent assessments of the applied implications of the research (e.g. [42]).

Because we have now shown stable effects with very brief tasks, with few learning trials, we predict that repeated playing of the game resulting in spaced learning could produce very stable effects when techniques are combined. For example, as well as the inhibition and speed of presentation used in the current study, other perceptual properties could be used, such as contrast/salience (e.g. [18,19,43]), smooth versus curved shape (e.g. [16,44]) or symmetry (e.g. [45]); as well as action-based fluency (e.g. [46]) that can be manipulated in tasks requiring direct reaching towards touch-screen phones and tablets. These techniques could be encountered independently and in combination as game players pursue goals such as gathering foods through various game levels.

Therefore, the approach we propose is that, as well as the explicit inhibition training techniques focused on overweight individuals engaged with dieting, a broader more implicit approach could also be taken. For example, the range of visuomotor techniques described above could be embedded within fun gaming environments, where there is no sense of overtly training, and as suggested here, often little awareness of game properties that affect preference. Such approaches could be effective with particular target populations, such as obese adults, but also more generally as a means of broadly improving health and well-being. In particular, embedding preference nudges within existing fun game environments might be especially effective for children across a wide age range, and such early interventions might provide long-term dividends.

Thus, it would perhaps be useful for these ideas to be considered in the future development of existing or new games to change behaviour. One approach could be to use converging methodologies where existing approaches based on social modelling such as the food dudes programme [47–49], could also use the basic visuomotor techniques discussed here in addition to the powerful techniques they already employ. Nudging behaviour simultaneously at these multiple levels could be more effective than any single approach. Hence, further basic science examining individual mechanisms and the combination of techniques, to identify in controlled experimental conditions which may or may not shift preferences, could be a valuable research programme to provide the foundations from basic research to real-world impact on behaviour change via gamification.

Ethics. All participants were recruited from the University of York's Department of Psychology participant recruitment system. Participants received either course credit or financial compensation for participation. Protocols were approved by the University of York's Psychology Departmental Ethics Committee and were in accord with the tenets of the Declaration of Helsinki. Participants gave written consent but were naive to the purpose of the research until participation was complete.

Data accessibility. Data, examples of stimuli, statistical models, supplementary analyses and documents are available at https://osf.io/vxbzj/.

Authors' contributions. B.M., J.C.F., H.O. and S.P.T. were involved in study conception and design. B.M. and J.C.F. were involved in data collection and data analysis. B.M., J.C.F., H.O. and S.P.T. were involved in manuscript draft and revision. All authors gave final approval for publication and agree to be held accountable for the work performed therein.

Competing interests. We declare we have no competing interests.

Funding. This research was supported by a Leverhulme Trust grant to S.P.T. and H.O. (grant no. RPG-2016-068).

Acknowledgements. We would like to thank Edward Hindmarsh and Fiona Day for assistance with data collection.

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
