## [Reviewer comments · Royal Society Open Science]

Review History

RSOS-200766.R0 (Original submission)

Review form: Reviewer 1

Is the manuscript scientifically sound in its present form?

Yes

Are the interpretations and conclusions justified by the results?

No

Is the language acceptable?

Yes

Do you have any ethical concerns with this paper?

No

Have you any concerns about statistical analyses in this paper?

No

Recommendation?

Accept with minor revision (please list in comments)

Comments to the Author(s)

Please see attached (Appendix A).

Decision letter (RSOS-200766.R0)

Dear Dr Flavell

On behalf of the Editors, we are pleased to inform you that your Manuscript RSOS-200766 "Three minutes to change preferences: Perceptual fluency and response inhibition" has been accepted for publication in Royal Society Open Science subject to minor revision in accordance with the referees' reports. Please find the referees' comments along with any feedback from the Editors below my signature.

Please submit your revised manuscript and required files (see below) no later than 7 days from today's (ie 27-Aug-2020) date. Note: the ScholarOne system will 'lock' if submission of the revision is attempted 7 or more days after the deadline. If you do not think you will be able to meet this deadline please contact the editorial office immediately.

on behalf of Dr Narayanan Srinivasan (Associate Editor) and Essi Viding (Subject Editor)
openscience@royalsociety.org

Associate Editor Comments to Author (Dr Narayanan Srinivasan):

Associate Editor: 1

Comments to the Author:

One expert reviewer has commented on the paper and raises a few issues that need to be addressed.

Reviewer comments to Author:

Reviewer: 1

Comments to the Author(s)

Please see attached.

===PREPARING YOUR MANUSCRIPT===

- one version identifying all the changes that have been made (for instance, in coloured highlight, in bold text, or tracked changes);
- a 'clean' version of the new manuscript that incorporates the changes made, but does not highlight them. This version will be used for typesetting.

===PREPARING YOUR REVISION IN SCHOLARONE===

Author's Response to Decision Letter for (RSOS-200766.R0)

See Appendix B.

Decision letter (RSOS-200766.R1)

Dear Dr Flavell,

It is a pleasure to accept your manuscript entitled "Three minutes to change preferences: Perceptual fluency and response inhibition" in its current form for publication in Royal Society Open Science.

on behalf of Dr Narayanan Srinivasan (Associate Editor) and Essi Viding (Subject Editor)
openscience@royalsociety.org

Appendix A

RSOS-200766: *Three minutes to change preferences: Perceptual fluency and response inhibition*

Evaluation:

The research reported here generally appears to have been well designed, the results appropriately analyzed, and there are no obvious problems that should prevent publication. The demonstration of preference change using a very-short procedure that may be suitable for gamification and potential application may be of interest to a relatively wide audience. However, there are issues related to the coverage of prior research, the rationale for specific choices of experimental techniques, and a key confound that limits interpretation of the results that all need to be addressed before the manuscript is suitable to publication.

Issues (in order of concern):

1. The use of a 'yucky' face as a stop signal makes it impossible to disentangle any effect of inhibition from well-established negative impact of pairing a neutral stimulus with an aversive stimulus (i.e., is the preference change due to affective consequences of inhibition or evaluative conditioning). The revisions to the manuscript need to acknowledge this confound and the fact that the 'inhibition' condition is actually a combination of inhibition + evaluative conditioning (which has also been used effectively to modify health behaviours—e.g., Lascelles et al., 2003)
2. Rationale for choice of stop-signal rather than Go/No-go inhibition. There is already a large literature in the health behaviours domain examining the effects of inhibition on food / alcohol preference and consumption using both stop-signal and Go/No-go tasks, and this work has shown the Go/No-go tasks are more effective at changing preference and behaviour because stimuli are associated with inhibition on all No-go trials, but on only a small subset of trials for stop-signal tasks. (see meta-analysis by Allom et al., 2016). It is therefore unclear why the authors would choose stop-signal in an attempt to show effects that survive shortened procedures and context changes. Including a bit in the introduction about the extensive work with Go/No-go (e.g., Kiss et al., 2008; Veling et al., 2008) and a bit more on the rationale for going instead with stop-signal is therefore warranted.
3. The challenge of interpreting the differences in the results of Exp. 1 and 2 is compounded by the fact that the authors changes so many things from one to the other. So it is impossible to know whether the differences are due to going from food to drink, the filler task, or the context change. Addressing this would be helpful in the interpretation of the results of Exp. 2. Providing a bit of rationale for why the authors changed so many things at once across experiments would also be helpful for readers.

4. The introduction overly emphasizes the uniqueness of the work for being the first to examine the effects of fluency and inhibition on preference change within the same experimental paradigm. Unfortunately, this overlooks the work by Fenske et al. (2004; they contrasted the effects of fluency via stimulus exposure duration and inhibition in a preview search task) and Frischen et al. (2012; they independently manipulated fluency via stimulus repetition and inhibition in a Go/No-go task). The combined effects of inhibition and other processes have also been examined in the realm of health behaviour, per se (e.g., Veling et al., 2014; inhibition and implementation intentions). The authors should consider this prior work and be a bit more cautious in their claims about the novelty of this aspect.
5. The innovative use of a 3-min procedure, on the other hand, probably deserves a bit more emphasis, as it truly is unique and noteworthy. Telling the readers about the shortest prior tasks to show effects of inhibition/fluency, rather than just telling about the upper limits on how long some prior studies have taken would be helpful in this regard.
6. P.5 – typos. ‘none’ should be ‘non’ in “...none laboratory...” and “...none game ...”

Appendix B

Dear Mr Dunn,

We are delighted for the very positive feedback and that our article is accepted for publication at RS.

We have made a number of changes, in line with the reviewer's comments. Reviewer comments are presented below in black and our responses in blue.

Yours sincerely,

Jonathan Flavell, Steven Tipper, Harriet Over, Bryony McKean

--o0o--

1. The use of a 'yucky' face as a stop signal makes it impossible to disentangle any effect of inhibition from well-established negative impact of pairing a neutral stimulus with an aversive stimulus (i.e., is the preference change due to affective consequences of inhibition or evaluative conditioning). The revisions to the manuscript need to acknowledge this confound and the fact that the 'inhibition' condition is actually a combination of inhibition + evaluative conditioning (which has also been used effectively to modify health behaviours—e.g., Lascelles et al., 2003)

We acknowledge the interesting point that the inhibition condition also possesses valence properties. This is made clear in the footnote 1 (page 12) in the paper, in the context of the work having an applied focus, and combined techniques will be more potent.

2. Rationale for choice of stop-signal rather than Go/No-go inhibition. There is already a large literature in the health behaviours domain examining the effects of inhibition on food / alcohol preference and consumption using both stop-signal and Go/No-go tasks, and this work has shown the Go/No-go tasks are more effective at changing preference and behaviour because stimuli are associated with inhibition on all No-go trials, but on only a small subset of trials for stop-signal tasks. (see meta-analysis by Allom et al., 2016). It is therefore unclear why the authors would choose stop-signal in an attempt to show effects that survive shortened procedures and context changes. Including a bit in the introduction about the extensive work with Go/No-go (e.g., Kiss et al., 2008; Veling et al., 2008) and a bit more on the rationale for going instead with stop-signal is therefore warranted.

The reviewer noted that go/no-go might be a more effective technique than stop-signal, and asked why we chose to employ the latter. In fact we had already explained why stop-signal was necessary in the MS in footnote 2 (page 12). We have now extended this footnote to make this issue clearer.

3. The challenge of interpreting the differences in the results of Exp. 1 and 2 is compounded by the fact that the authors changes so many things from one to the other. So it is impossible to know whether the differences are due to going from

food to drink, the filler task, or the context change. Addressing this would be helpful in the interpretation of the results of Exp. 2. Providing a bit of rationale for why the authors changed so many things at once across experiments would also be helpful for readers.

The changes between experiments 1 and 2 are indeed substantial. We make clear that this was to produce a very strong challenge for the effects we are investigating. It was essential to create situations that more closely matched the real-world, such as actual food consumption and stability in memory across changed contexts. We have added further comments, and think the value of this will be clearer to readers.

4. The introduction overly emphasizes the uniqueness of the work for being the first to examine the effects of fluency and inhibition on preference change within the same experimental paradigm. Unfortunately, this overlooks the work by Fenske et al. (2004; they contrasted the effects of fluency via stimulus exposure duration and inhibition in a preview search task) and Frischen et al. (2012; they independently manipulated fluency via stimulus repetition and inhibition in a Go/No-go task). The combined effects of inhibition and other processes have also been examined in the realm of health behaviour, per se (e.g., Veling et al., 2014; inhibition and implementation intentions). The authors should consider this prior work and be a bit more cautious in their claims about the novelty of this aspect.

We further note other work that has also investigated perceptual fluency and inhibition processes within the same research programme.

5. The innovative use of a 3-min procedure, on the other hand, probably deserves a bit more emphasis, as it truly is unique and noteworthy. Telling the readers about the shortest prior tasks to show effects of inhibition/fluency, rather than just telling about the upper limits on how long some prior studies have taken would be helpful in this regard.

We appreciate that the reviewer felt the 3 minute intervention was unique and important. We have added some comparisons to existing experiment trial lengths as durations are not often reported.

6. P.5 - typos. 'none' should be 'non' in "...none laboratory..." and "...none game ..."

Typos corrected.